# Monocentric observational cohort study to investigate the transmission of third-generation cephalosporin-resistant Enterobacterales in a neonatal intensive care unit in Heidelberg, Germany

Dennis Nurjadi,[1,2,3] Vanessa M. Eichel,[2] Johannes Pöschl,[4] Christian Gille,[4] Simon Kranig,[4] Klaus Heeg,[2] Sébastien Boutin[1,2]

**ABSTRACT** Third-generation cephalosporin-resistant Enterobacterales is a major threat for newborns in neonatal intensive care units (NICUs). The route of acquisition in a non-outbreak setting should be investigated to implement adequate infection prevention measures. To identify risk factors for colonization with and to investigate the transmission pattern of third-generation cephalosporin-resistant Enterobacterales in a NICU setting. This monocentric observational cohort study in a tertiary NICU in Heidelberg, Germany, enrolled all hospitalized neonates screened for cephalosporin-resistant Enterobacterales. Data were collected from 1 January 2018 to 31 December 2021. Weekly screening by rectal swabs for colonization with third-generation cephalosporin-resistant Enterobacterales was performed for all newborns until discharge. Whole-genome sequencing was performed for molecular characterization and transmission analysis. In total, 1,287 newborns were enrolled. The median length of stay was 20 (range 1–250) days. Eighy-eight infants (6.8%) were colonized with third-generation cephalosporin-resistant Enterobacterales. Low birth weight [<1500 g (adjusted odds ratio, 5.1; 95% CI 2.2–11.5; $P < 0.001$)] and longer hospitalization [per 30 days (adjusted odds ratio, 1.7; 95% CI 1.5–2.0; $P < 0.001$)] were associated with colonization or infection with drug-resistant Enterobacterales in a multivariate analysis. *Enterobacter cloacae* complex was the most prevalent third-generation cephalosporin-resistant Enterobacterales detected, 64.8% (59 of 91). Whole-genome sequencing, performed for the available 85 of 91 isolates, indicated 12 transmission clusters involving 37 patients. This cohort study suggests that transmissions of third-generation cephalosporin-resistant Enterobacterales in newborns occur frequently in a non-outbreak NICU setting, highlighting the importance of surveillance and preventive measures in this vulnerable patient group.

**IMPORTANCE** Preterm newborns are prone to infections. Therefore, infection prevention should be prioritized in this vulnerable patient group. However, outbreaks involving drug-resistant bacteria, such as third-generation resistant Enterobacterales, are often reported. Our study aims to investigate transmission and risk factors for acquiring third-generation cephalosporin-resistant Enterobacterales in a non-outbreak NICU setting. Our data indicated that premature birth and low birth weight are significant risk factors for colonization/infection with third-generation cephalosporin-resistant Enterobacterales. Furthermore, we could identify putative transmission clusters by whole-genome sequencing, highlighting the importance of preemptive measures to prevent infections in this patient collective.

Address correspondence to Dennis Nurjadi, dennis.nurjadi@uni-luebeck.de.

The authors declare no conflict of interest.

**KEYWORDS** cephalosporin-resistant Enterobacterales, whole-genome sequencing, infection prevention and control, neonatology, antimicrobial resistance, transmission, ICU

Neonatal infections and sepsis are among the leading causes of mortality in newborns (1–3). Especially premature infants and those with low birth weight are vulnerable to infections and sepsis (4, 5). Although antibiotics are available for treatment, the emergence of third-generation cephalosporin-resistant Enterobacterales is an imminent threat to be taken seriously (6, 7). Due to the clinical impact of such bloodstream infections and sepsis on this vulnerable patient group, emphasis is set on preventing transmission of colonization and infection with third-generation cephalosporin-resistant Enterobacterales (8). Regular screening regimens have been implemented in Germany in 2013, even in non-outbreak settings, aiming to anticipate and timely detect the emergence of these drug-resistant pathogens (2, 9, 10).

Thus, active screening and surveillance measures are usually part of the standard infection prevention and control measures in most neonatal intensive care units (NICUs) in Germany. However, molecular characterization by whole genome sequencing (WGS) is typically only performed on sporadic occasions or upon outbreak suspicion. In most cases, the prevalence of third-generation cephalosporin-resistant Enterobacterales in this patient group is considerably low. Furthermore, a significant proportion is acquired via maternal-neonatal transmission, questioning the necessity of constantly monitoring of patient-to-patient transmissions (11, 12). Nonetheless, outbreaks of drug-resistant Enterobacterales have been reported numerously and should be prevented.

The current gold standard for the molecular typing of bacteria is WGS. It has been demonstrated on numerous occasions that integrating of WGS is beneficial in elucidating and tracing back outbreak scenarios and transmission events (13, 14). Although the costs of performing WGS have dropped over the last few years, there is still some reluctance to integrate WGS for active and systematic surveillance in clinical practice. Over the past years, weekly screening has been implemented in the NICU of our tertiary hospital to screen hospitalized newborns for colonization with *Staphylococcus aureus*, third-generation cephalosporin-resistant and carbapenem-resistant Enterobacterales since colonization with these bacteria is associated with an increased risk of acquiring infections (15). However, molecular typing was only performed upon suspicion due to the accumulation of detection or certain antibiotic susceptibility patterns (16). Therefore, we wonder whether our surveillance approach adequately identifies transmission events and whether systematic WGS (i.e., performing WGS on all isolates) would improve our surveillance measures.

To answer this question, we conducted an observational cohort study to identify clinical characteristics and parameters associated with third-generation cephalosporin-resistant Enterobacterales colonization. Further, we sequenced all available third-generation cephalosporin-resistant Enterobacterales isolated at our NICU over 4 years between 1 January 2018 and 31 December 2021 to investigate the number and magnitude of transmission events as a potential acquisition route for colonization with third-generation cephalosporin-resistant Enterobacterales in this vulnerable patient group.

## RESULTS

In total, 1,287 newborns were treated in the neonatal intensive care unit of our tertiary care hospital. The baseline characteristics of the study population are summarized and displayed in Table 1. Overall, the mean length of hospitalization was 34.6 days (range 0–355 days). The median time to first detection of third-generation cephalosporin-resistant Enterobacterales was 24 days (range 3–150 days). Colonization and infection with third-generation cephalosporin-resistant Enterobacterales were detected in 88 of 1,287 (6.8 %) newborns and were associated with higher odds for neonatal sepsis (OR = 6.6, 95% CI, 2.9–15.0), $P < 0.001$) and longer hospitalization (median 79 days, IQR, 19–201 days versus 18 days, IQR, 1–194 days). Newborn sepsis is defined according

**TABLE 1** Baseline and clinical characteristics of the study population[g,h]

| | Total[a] | Third-generation cephalosporin-resistant Enterobacterales[a] | | Crude[b] | | Adjusted[c] | |
|---|---|---|---|---|---|---|---|
| | | yes, n = 88 | n = 1199 | OR (95% CI) | P | OR (95% CI) | P |
| | n (%) | n (%) | n | | | | |
| Female | 577 (44.8) | 42 (47.7) | 535 (44.6) | 1.1 (1.7–1.8) | .6 | NA | NA |
| Gestation age <37 weeks | 982 (76.3) | 80 (90.9) | 902 (75.3) | 3.3 (1.6–6.9) | .002 | 0.5 (0.2–1.4) | 0.2 |
| Birth weight <1500 g | 407 (31.6) | 71 (80.7) | 336 (28.0) | 10.7 (6.2–18.5) | <0.001 | 5.1 (2.2–11.5) | <0.001 |
| Cesarean delivery[d] | 903 (70.5) | 72 (82.8) | 831(69.6) | 2.1 (1.2–3.7) | .01 | 1.4 (0.7–2.6) | 0.4 |
| Multiple gestations | 297 (23.1) | 26 (29.9) | 271 (22.6) | 1.5 (0.9–2.4) | .1 | NA | NA |
| Sepsis[e] | 30 (2.4) | 9 (10.8) | 21 (1.8) | 6.6 (2.9–15.0) | <0.001 | NA | NA |
| Antibiotic exposure[f] | 982 (77.4) | 74 (86.1) | 908 (76.8) | 1.8 (1.0–3.5) | .05 | 1.1 (0.6–2.2) | 0.8 |
| LOS (per 30 days)[d] | 20 (1–250) | 79 (19–201) | 18 (1–194) | 2.1 (1.3–2.4)[d] | <0.001 | 1.7 (1.5–2.0) | <0.001 |

[a]Unless otherwise indicated, data are expressed as number (%) of patients.
[b]Ratio of odds of rectal colonization, calculated using a univariate logistic regression model.
[c]Ratio of odds of rectal colonization, calculated using a multivariate logistic regression model, with birthweight, gestation age, delivery mode, antibiotic exposure and length of stay (per 30 days). Mean inflation factor was 1.41 (range 1.04–1.89), indicating no collinearity.
[d]Six missing data; 1 from colonized and 5 from non-colonized newborns.
[e]Thirty-seven missing data; 5 from colonized and 32 from non-colonized newborns; sepsis categorization based on clinical parameters/symptoms.
[f]Nineteen missing data; 2 from colonized and 17 from non-colonized newborns.
[g]One missing data from 1 colonized newborn, odds ratio calculated per 30 days hospitalization.
[h]OR, odds ratio; LOS, length of stay; NA, not applicable.

to clinical and laboratory parameters as defined by the national infection surveillance system (NEO KISS) (17). In our study period, 9 of 88 (10.2%) colonized newborns had an infection with third-generation cephalosporin-resistant Enterobacterales, of which six were bloodstream infections, two were meningitis, and one was wound infection. All newborns recovered from their infections.

Premature (<37 gestation weeks) birth (OR = 3.3, 95% CI 1.6–6.9, P = 0.002), birthweight of <1500 g (OR = 10.7, 95% CI 6.2–18.5, P < 0.001), and delivery via cesarean section (OR = 2.1, 95% CI 1.2–3.7, P = 0.01) were significant factors associated with colonization with third-generation cephalosporin-resistant Enterobacterales in our study population. In a multivariate logistic regression analysis, birth weight under 1,500 g and length of stay were the independent and strongest effect modifiers (adjusted OR = 5.1, 95% CI 2.2–11.5, P < 0.0.001 and adjusted OR for length of stay per 30 days = 1.7, 95% CI 1.5–2.0, P < 0.001, Table 1).

## Bacterial species identified

Overall, 91 Enterobacterales isolates from 88 patients were collected throughout the 4-year study period. Only the first detection per patient per species was considered for the end analysis. Three patients (3/88, 3.4%) were colonized with two different third-generation cephalosporin-resistant Enterobacterales. The most common Enterobacterales isolated was *Enterobacter cloacae* complex (59/91, 64.8%), followed by *Citrobacter freundii* (10/91, 11.0%), *Klebsiella aerogenes* (9/91, 9.9%), *Escherichia coli* (7/91, 7.7%), *Serratia marcescens* (5/91, 5.5%), and *Citrobacter braakii* (1/91, 1.1%). Eight-nine of ninety-two bacterial isolates were able to be recovered for WGS (3/59 *E. cloacae* complex were non-recoverable, Fig. 1). The mechanism of third-generation cephalosporin resistance was mainly intrinsic for *E. cloacae* complex, *K. aerogenes*, *S. marcescens,* and *Citrobacter* sp. Only the *E. coli* isolates harbored mobile resistance genes encoding for extended-spectrum beta-lactamases ($bla_{CTX-M-15}$). An overview of the resistome (resistance genes detected from WGS) is provided in Fig. S1.

## Epidemiology and transmission

The transmission analysis based on SNP was performed and presented for each species, respectively. Transmission clusters are defined as isolates with ≤16 SNPs for *E. hormaechei*, 2 for *C. freundii*, 19 for *K. aerogenes,* and 4 for *E. coli* based on the method of Duval

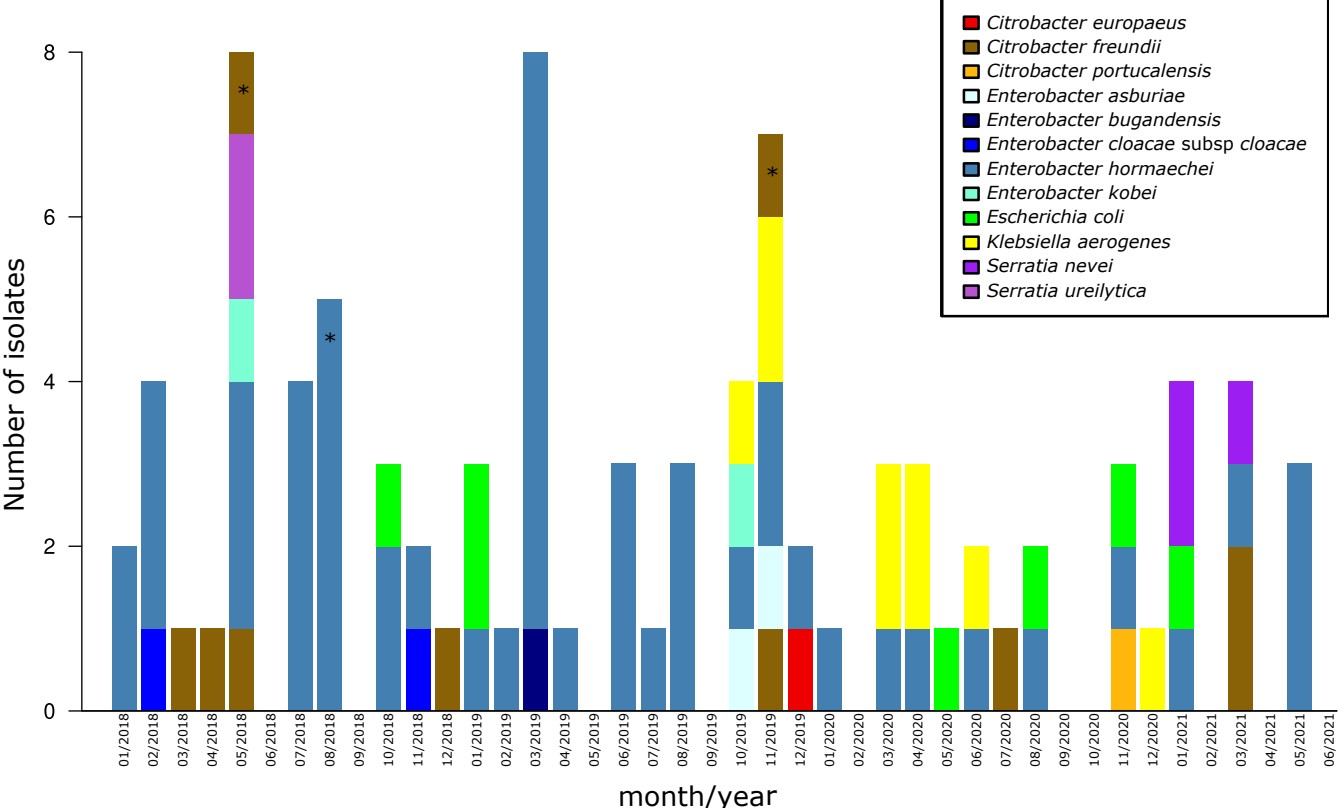

**FIG 1** Overview of third-generation cephalosporin-resistant Enterobacterales included in the study. Non-recoverable isolates are indicated by an asterisk (*).

et al. (18) Overall, we identified 12 putative transmission clusters involving 37 newborns across the various bacterial species. The largest clusters consisted of 13 and 5 newborns, whereas the other clusters only consisted of 2–3 newborns. The hospital hygiene only detected the two largest clusters and failed to identify all 13 patients as part of the most prominent transmission cluster (10 patients were correctly classified and 3 patients were misclassified).

### Enterobacter cloacae complex

The epidemiological curve suggested that there is an overrepresentation of *Enterobacter hormaechei*, suggesting that a potential outbreak or at least multiple transmission events (Fig. 1). Of 59 isolates, 56 were sequenced. Identification based on the assembled genomes revealed that most isolates were *Enterobacter hormaechei* (n = 51), followed by *Enterobacter asburiae*, *Enterobacter cloacae* subspecies *cloacae*, *Enterobacter kobei* (n = 2 each), and *Enterobacter bugadensis* (n = 1). The phylogeny of *E. hormaechei,* along with detected AMR genes, plasmids, and ward occupancy are displayed in Fig. 2. Resistance toward third-generation cephalosporin was mainly mediated by AmpC ($bla_{ACT}$). We identified four potential transmission clusters. Two patients are involved in transmission cluster A (ST158) with zero SNP between isolates. Cluster B (ST419) consists likewise of two patients with zero SNP between both isolates. Cluster C (ST78) is the largest cluster involving 13 patients with 0–15 SNPs between isolates. The last transmission cluster D (ST295) consists of isolates from two patients with three SNP difference between the isolates. The transmission cluster as defined by WGS correlates with the spatiotemporal overlap in the ward occupancy. Both patients in cluster A overlapped by 211 days, in cluster B overlapped by 81 days, and in cluster D with 79 days overlap. For cluster C, not all patients overlap with each other. The first three patients (which were not identified as potential outbreak via the traditional screening) and the last 10 patients (in chronological

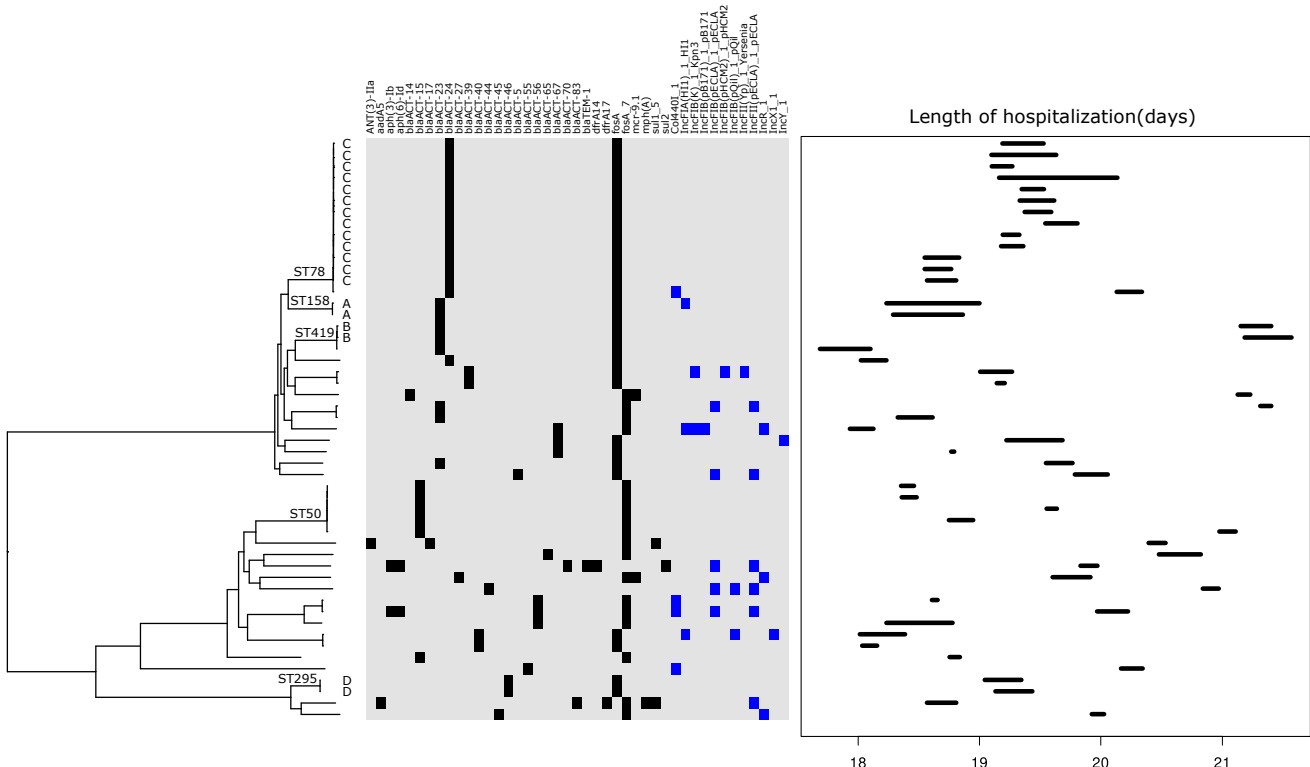

**FIG 2** Phylogenetic analysis of *Enterobacter hormaechei*. Isolates belonging to the same clonal type are coded with an identical letter on the tree. The presence of antimicrobial resistance genes is symbolized with a black square while replicon genes indicating the plasmid type are indicated by a blue square. On the left panel, the patient's hospital stay is represented in a Gantt diagram. Patients are ordered according to the position of their isolate on the phylogenetic tree.

order) had overlapping ward occupancy ranging from 6 to 173 days, but the two groups did not share a spatiotemporal overlap. In patients colonized by *E. hormaechei*, one sibling pair was colonized by the same clone (in cluster C), but two other twins were both colonized by different clones of *E. hormaechei*. For *E. asburiae*, both isolates were closely related with three SNPs (ST657) so that transmission was very likely. There were no putative transmissions for both *E. cloacae* subspecies *cloacae* and *E. kobei*.

## Citrobacter freundii

For *C. freundii* (*n* = 8), WGS analysis indicated a transmission event involving two patients (ST98) with two SNPs between isolates (Fig. 3A, cluster A). Both patients stayed in the same ward with a time overlap of 113 days, thus suggesting that a transmission event was likely and plausible. We detected one *Citrobacter europaeus* and one *Citrobacter portucalensis*. All *C. freundii* in our collection harbor various variants of the cephamycinase gene, $bla_{CMY}$ (Fig. 3A).

## Klebsiella aerogenes

Altogether, we isolated nine *K. aerogenes* in our study period. WGS analysis suggested two putative transmission clusters; involving two patients (five SNPs between isolates) in the first cluster (cluster A) and five patients in cluster B with 5–15 SNPs between the isolates within the cluster. In Cluster A, both patients had 42 days of overlap in hospitalization in the same ward. In Cluster B, the ward occupancy overlapped between 32 and 66 days (Fig. 3B). In all the *K. aerogenes* isolates, the chromosomal AmpC ($bla_{ACT}$) was the most likely mediator of cephalosporin resistance.

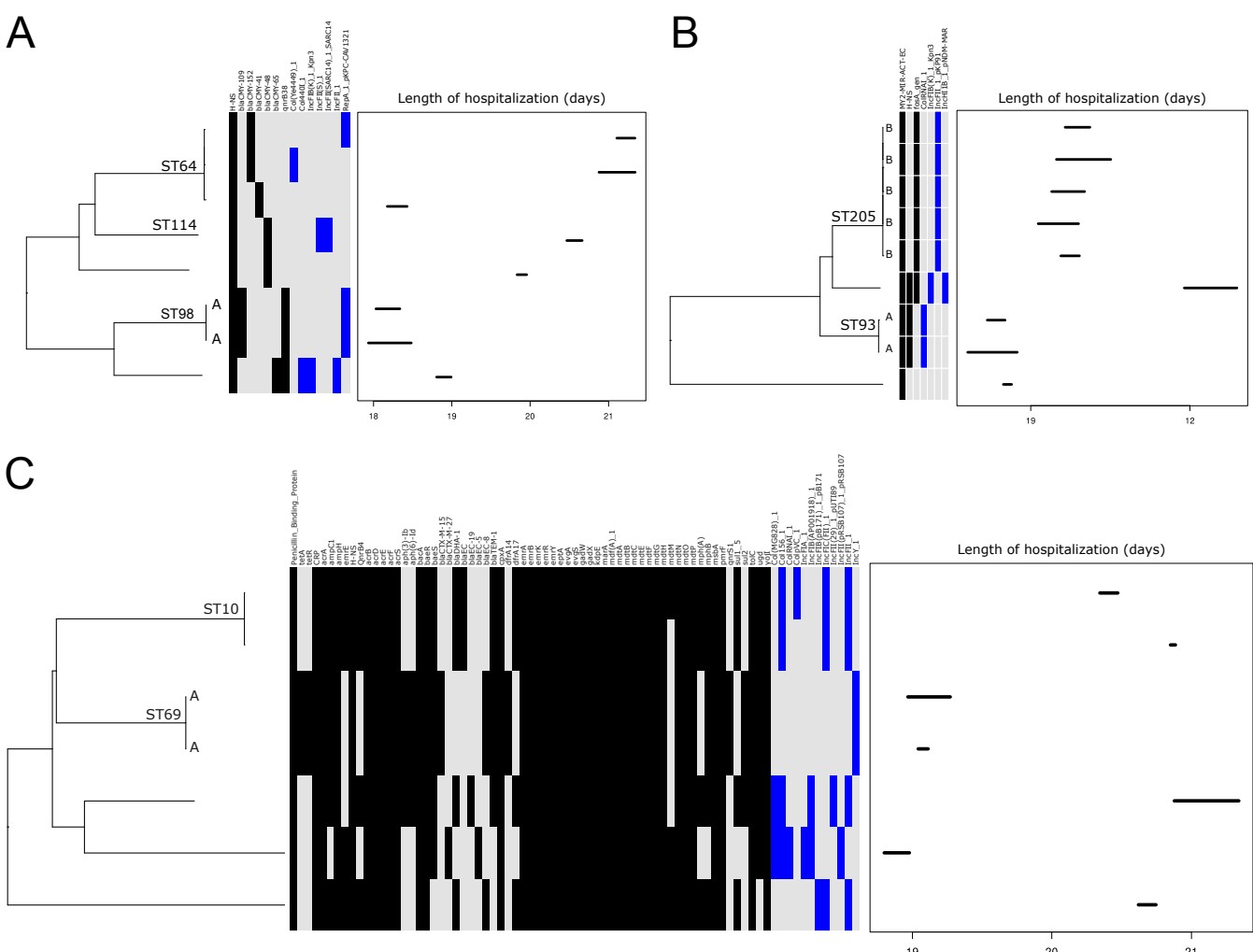

**FIG 3** Phylogenetic analysis of (A) *Citrobacter freundii,* (B) *Klebsiella aerogenes,* and (C) *Escherichia coli.* Isolates belonging to the same clonal type are coded with an identical letter on the tree. The presence of antimicrobial resistance genes is symbolized with a black square while replicon genes indicating the plasmid type are indicated by a blue square. On the left panel, the patient's hospital stay is represented in a Gantt diagram. Patients are ordered according to the position of their isolate on the phylogenetic tree.

## Escherichia coli

Of the seven patients with third generation cephalosporin-resistant *E. coli*, two patients were involved in a transmission cluster of ST69 *E. coli* harboring $bla_{\text{CTX-M-15}}$ and $bla_{\text{TEM-1}}$ with four SNPs between isolates. There were 27 days of overlap in ward occupancy (Fig. 3C). None of the patients have siblings.

## Serratia species

We detected three patients with *Serratia nevei* and two patients with *Serratia ureilytica*. The close genetic relatedness of all three *S. nevei* isolates with two to four SNPs indicated a transmission cluster. There were 74–105 days of overlap in ward occupancy. Similarly, the two *S. ureilytica* isolates were also closely related with six SNPs between them. Both patients with *S. ureilytica* stayed in the same ward with 30 days overlap.

## DISCUSSION

Hospitalized newborns, especially those with low birth weight (premature) are vulnerable to infections due to various nosocomial pathogens, which correlates with

high morbidity, longer hospitalization, and mortality. The most relevant and dreaded pathogens associated with neonatal sepsis is *Staphylococcus aureus* and gram-negative rods (15, 19, 20). Therefore, the German national guidelines for hospital hygiene and infection prevention (KRINKO) recommends regular screening as means of active surveillance to timely identify potential transmission chains for *S. aureus* and drug-resistant (cephalosporin- and carbapenem-resistant) gram-negative rods. However, active molecular surveillance is not the current gold standard in the clinical routine. Due to the high cost, low prevalence of third-generation cephalosporin-resistant Enterobacterales and considerably large amount of efforts and resources needed to implement active molecular surveillance using WGS, the cost-effectiveness is much debated (21, 22).

In our study, we identified 12 transmission clusters involving 37 newborns over a period of 4 years. Although two transmission clusters involved a significant number of patients (ST78 *E. hormaechei* with 13 patients and *K. aerogenes* with 5 patients), most transmission clusters involved only two to three patients and were therefore difficult to detect. Indeed, the hospital hygiene team only identified the two largest clusters as outbreak clusters based on routine microbiology results and epidemiological overlap. Even so, in the largest cluster involving 13 patients, only 10 of the 13 patients were identified as a potential outbreak cluster since the first 3 patients (in chronological order of detection) did not have any spatiotemporal overlap with the remaining 10 patients. Therefore, our study demonstrated that small transmission events involving only two to three patients were missed by the current standard of practice. In the case of the ST78 *E. hormaechei* outbreak, timely initiation of infection prevention and control measures may have prevented this outbreak from reaching such a magnitude.

For effective infection prevention and control measures, it is important to understand the transmission route and dynamics. In the NICU, direct patient-to-patient transmission is highly unlikely since the mobility of newborns is limited. Other potential routes of transmission are through sharing of equipment, environmental contamination, and healthcare worker-mediated transmissions (23, 24). In the NICU, especially for preterm newborns, skin-to-skin contact through kangaroo care is encouraged so that kangaroo care should be considered as potential routes of acquisition for third-generation cephalosporin-resistant Enterobacterales (25).

In line with other studies, preterm, low birth weight, and length of stay are associated with third-generation cephalosporin-resistant Enterobacterales positivity in our study (26–28). Interestingly, cesarean delivery was also significantly associated with third-generation cephalosporin-resistant Enterobacterales positivity. The colonization of the vaginal tract with extended-spectrum beta-lactamase producing *Enterobacteriaceae* (ESBL) has been demonstrated to be a significant risk factor for infant colonization with ESBL (27–29). Antibiotic prophylaxis during cesarean section or post-natal antibiotic therapy may promote the selection of third-generation cephalosporin-resistant Enterobacterales in both the mother and the newborn (28). The most abundant species in our isolate collection were *E. cloacae* complex and *K. aerogenes*. Both species harbor the inducible chromosomal *ampC* gene, which can be upregulated upon exposure with cephalosporins. Thus, exhibiting third-generation cephalosporin-resistant phenotype and hence providing a plausible explanation for the abundance of these species (16, 30). Only a minor proportion of isolates harbor an ESBL gene, such as $bla_{CTX-M}$, as a resistance mechanism, so antibiotic use may be an important key driver of third-generation cephalosporin-resistant Enterobacterales in this patient population. Furthermore, sepsis is associated with third-generation cephalosporin-resistant Enterobacterales positivity in this study. This observation lends support to the hypothesis on the emergence of third-generation cephalosporin-resistant Enterobacterales due to antibiotic exposure since empirical therapy with antibiotics is often administered for suspected neonatal sepsis based on risk factors and clinical assessment (31).

Our study has some limitations. In microbiological diagnostics, only one representative colony is picked for species identification and antibiotic susceptibility testing, so this

algorithm does not consider a heterogeneous population of the same species; hence, random colony picking may have underestimated the magnitude of transmission events. In the study, we did not include isolates from environmental sampling, the parents, and healthcare workers so that a complete analysis to elucidate the transmission route is not possible. Nevertheless, we could demonstrate that incorporating WGS into the active surveillance measures can detect silent transmissions, which may improve infection prevention strategies.

This cohort study could confirm published data on risk factors for colonization and infection with third-generation cephalosporin-resistant Enterobacterales. Our data indicate that colonization with third-generation cephalosporin-resistant Enterobacterales is associated with increased odds of the clinical presentation of neonatal sepsis. Furthermore, molecular characterization using WGS revealed that transmission between patients could occur and remain undetected, highlighting the importance of systematic surveillance to guide infection prevention measures in this vulnerable patient group.

## MATERIALS AND METHODS

### Study population and study setting

For this cohort study, we enrolled all newborns admitted to the NICU of the Heidelberg University Hospital from 1 January 2018 to 31 December 2021. The NICU is equipped with 24 stationary beds, assembled as two 4-bed rooms and seven 2-bed rooms. The primary endpoint of this observational cohort study was colonization or infection with third-generation cephalosporin-resistant Enterobacterales during hospitalization. Secondary outcomes were the transmission of third-generation cephalosporin-resistant Enterobacterales in a non-outbreak setting and the risk factors of acquiring third-generation cephalosporin-resistant Enterobacterales during hospitalization.

Weekly rectal screenings of newborns for third-generation cephalosporin-resistant Enterobacterales were conducted from the time of birth to hospital discharge as part of the local infection control and screening policy for multidrug-resistant organisms in concordance with national recommendations from the Commission for Hospital Hygiene and Infection Prevention (KRINKO) of the Robert Koch Institute. The local ethics committee was consulted prior to study begin and waived individual informed consent owing to de-identified data (S474/2018). We adhered to the Strengthening the Reporting of Observational Studies in Epidemiology (STROBE) guideline in reporting the findings of our observational cohort study, wherever applicable.

### Screening and infection prevention procedures

Screening and infection prevention procedures were performed according to the national guideline (10). Rectal/perianal screening using eSwab (Copan) for colonization with a third-generation cephalosporin-resistant gram-negative bacilli was performed weekly in the postnatal period. Basic hygiene measures were applied for any patient contact. These measures included consistent hand disinfection in accordance with the five World Health Organization indications and wearing disposable gloves and protective gowns to avoid contamination of staff where direct contact with blood, secretions, excrement, mucous membranes, or non-intact skin is expected.

Furthermore, the following infection prevention and control measures were implemented: (i) disposable apron for nursing rounds and in case of contamination risk of the body front; (ii) protective gown, if the child is carried; (iii) regular and hygiene training and compliance observations (at least once a year), on-site visits (at least once a year), and quality meetings with the infection control team (at least three times a year); (iv) 24 hours monitoring of automated hand disinfectant use at every bed site; (v) surgical face mask for the care of patients with methicillin-susceptible *S. aureus* (MSSA) or methicillin-resistant *S. aureus* (MRSA) to avoid droplet contamination; and (vi) isolation room and contact precautions for patients with MRSA or third-generation cephalosporin-resistant or carbapenem-resistant gram-negative bacteria.

## Microbiological diagnostics

Specimens from weekly screenings were processed in the microbiological diagnostics laboratory. Swabs were inoculated onto a chromogenic selective medium for ESBL (extended-spectrum beta-lactamase-producing gram-negative rods; ChromIDESBL, Biomérieux) and BD Columbia blood agar with 5% sheep blood (BD Diagnostics) as a growth control for sampling validity. Bacterial species identification was performed via matrix-assisted laser desorption ionization mass spectrometry (MALDI-TOF MS; Bruker). Antibiotic susceptibility testing was performed using VITEK2 (Biomérieux) and interpreted according to the EUCAST guidelines in the respective year. A change in the interpretation antibiotic susceptibility testing was implemented in 2019 according to the re-classification of "intermediate susceptibility" to "susceptible at higher exposure." For this study, resistance to third-generation cephalosporins (ceftriaxone/cefotaxime) was defined as "resistant" according to the respective clinical breakpoints, "intermediate" were considered as "susceptible" isolates. Bacterial isolates were cryopreserved at −70°C for molecular typing.

## Whole genome sequencing and data analysis

DNA extraction, library preparation, sequencing on a MiSeq Illumina platform (short-read sequencing, 2 × 300 bp) and post-sequencing procedure were performed as previously described (32). Briefly, raw sequences were controlled for quality using fastp (v0·23·2 with parameters -q = 30 and -l = 45) and assembled with SPAdes 3.15.5 (with the option—careful and—only-assembler) (33, 34). Draft genomes were curated by removing contigs with a length <500 bp and/or coverage <10×. The quality of the final draft was quality-controlled using Quast (v5.0.2) (35). The complete draft genomes were processed through available databases using Abricate (https://github.com/tseemann/abricate) to identify antimicrobial resistance (NCBI, CARD, ARG-ANNOT, ResFinder, and MEGARES databases) and plasmid type (PlasmidFinder database) to identify the Inc type of the plasmid (36, 37). The species identification of each draft genome was done using mash (sub-command screen) by screening each draft genome to a database composed of a representative genome of each species present in the Microbial Genomes resource (https://www.ncbi.nlm.nih.gov/genome/microbes/). Furthermore, each draft genome was aligned to its representative genome reference from the Microbial Genomes resource using SKA. The alignment was then analyzed with Gubbins 3.2.1 to define SNPs distance, and phylogenetic relationship was determined using the R package samestrains following the methodology of Duval et al. (18, 38).

## Statistical analysis

We used univariable and multivariable logistic regression to estimate the change in the odds of colonization with third-generation cephalosporin-resistant Enterobacterales in the presence of putative risk factors of these outcomes, together with their 95% CIs, and tested against the null hypothesis (H0) with an odds ratio of 1.00 using an α of 0.05. Test of the collinearity for correlations of the variables was performed using the (variance inflation factor) command following a regression model with all relevant variables. The mean of the variance inflation factor was 1.84 (range 1.08–3.48), indicating no significant collinearity between variables. All statistics were performed in STATA, v17 (StataCorp LLC).

## ACKNOWLEDGMENTS

Institutional funding.

## AUTHOR AFFILIATIONS

[1]Department of Infectious Diseases and Microbiology, University of Lübeck, Lübeck, Germany

²Department of Medical Microbiology and Hygiene, Heidelberg University Hospital, Heidelberg, Germany

³German Center for Infection Research (DZIF), Partner Site Hamburg-Lübeck-Borstel-Riems, Lübeck, Germany

⁴Department of Neonatology, Heidelberg University Hospital, Heidelberg University Children's Hospital, Heidelberg, Germany

## AUTHOR ORCIDs

Dennis Nurjadi  http://orcid.org/0000-0002-1278-5939
Sébastien Boutin  http://orcid.org/0000-0002-0499-2460

## AUTHOR CONTRIBUTIONS

Dennis Nurjadi, Conceptualization, Formal analysis, Investigation, Methodology, Project administration, Supervision, Validation, Writing – original draft, Writing – review and editing | Vanessa M. Eichel, Data curation, Investigation, Writing – original draft, Writing – review and editing | Johannes Pöschl, Data curation, Investigation, Methodology, Resources, Writing – original draft, Writing – review and editing | Christian Gille, Data curation, Investigation, Methodology, Writing – original draft, Writing – review and editing | Simon Kranig, Data curation, Investigation, Resources, Writing – original draft, Writing – review and editing | Klaus Heeg, Conceptualization, Investigation, Methodology, Resources, Supervision, Writing – original draft, Writing – review and editing | Sébastien Boutin, Conceptualization, Data curation, Formal analysis, Investigation, Methodology, Supervision, Visualization, Writing – original draft, Writing – review and editing

## DATA AVAILABILITY

Sequencing data are available in the NCBI repository under the Bioproject number PRJNA954276. Accession numbers of the isolates are included in the Supplementary Data.

## ADDITIONAL FILES

The following material is available online.

### Supplemental Material

**Supplementary Appendix (Spectrum02038-23-s0001.docx).** Supplemental methods, Table S1, and Fig. S1.

### Open Peer Review

**PEER REVIEW HISTORY (review-history.pdf).** An accounting of the reviewer comments and feedback.

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
