## [Reviewer comments · Microbiology Spectrum]

Microbiology Spectrum

Monocentric observational cohort study to investigate the transmission of third-generation cephalosporin-resistant Enterobacterales in a neonatal intensive care unit, Heidelberg, Germany.

Dennis Nurjadi, vanessa eichel, Johannes Pöschl, Christian Gille, Simon Kranig, Klaus Heeg, and Sébastien Boutin

Corresponding Author(s): Dennis Nurjadi, Universitat zu Lubeck

Review Timeline:

Submission Date:	May 15, 2023
Editorial Decision:	June 26, 2023
Revision Received:	July 25, 2023
Editorial Decision:	July 31, 2023
Revision Received:	July 31, 2023
Accepted:	August 4, 2023

Editor: Katharina Schaufler

Reviewer(s): Disclosure of reviewer identity is with reference to reviewer comments included in decision letter(s). The following individuals involved in review of your submission have agreed to reveal their identity: Ulises Garza-Ramos (Reviewer #1)

Transaction Report:

DOI: <https://doi.org/10.1128/spectrum.02038-23>

June 26, 2023

Prof. Dennis Nurjadi
Universität zu Lubeck
Department of Infectious Diseases and Microbiology
Ratzeburger Allee 160 Haus 50
Lübeck 23538
Germany

Re: Spectrum02038-23 (Colonization, transmission, and molecular characterization of third-generation cephalosporin-resistant Enterobacteriaceae in a neonatal intensive care unit, 2018-2021, Germany.)

Dear Prof. Dennis Nurjadi:

Link Not Available

Sincerely,

Katharina Schaufler

Journals Department
Reviewer comments:

Reviewer #1 (Comments for the Author):

The study describes the characterization of colonizing isolates resistant to third-generation cephalosporins in a neonatal ICU.

Around the document they erroneously describe "third-generation cephalosporin Enterobacterales"

The title should be modified by eliminating the study period; including the country. That must be very well specified in M and M.

The bacteria are isolated from the rectum of children; explaining colonization; however, whether there were infections in children is not described. It is not disputed how much it affected colonization and thus the results of the prevention measures taken during the study; in the same way the possible infections.

It is described that the identification by MALDI-TOF was used, and the results are not shown and if they agree with the species identified by WGS.

The antimicrobial susceptibility obtained with VITEK2 does not correlate with the resistome obtained.

The ideal way to identify bacterial species is the average nucleotide identity analysis; Therefore, it is suggested to use this program for the different species and correlate them with the MALDI-TOF result; Since if the MALDI-TOF database is not updated, it does not specifically identify the species of complexes.

Introduction; detail more other studies from your hospital and what has changed.

Results.

Line 95-96, colonization or infection; it is not clear.

It would be necessary to describe and more to discuss the interesting data of figure 1. Why these species and how was it during the time?

The Supplementary Figure S1 should be more described in results and correlate with ST and something that is not discussed and determined are the incompatibility groups of the plasmids. See an example. Duran-Bedolla J, Rodríguez-Medina N, Dunn M, Mosqueda-García D, Barrios-Camacho H, Aguilar-Vera A, Aguilar-Vera E, Suárez-Rodríguez R, Ramírez-Trujillo JA, Garza-Ramos U. Plasmids of the incompatibility group FIBK occurs in *Klebsiella variicola* from diverse ecological niches. *Int Microbiol*. 2023 Mar 27. doi: 10.1007/s10123-023-00346-0. Epub ahead of print. PMID: 36971854.

Little transmission of bacteria is identified, what is not clear is the phylogenetic relationship (clones) between the identified bacteria that were transmitted. This makes sense to the discussion that the measures taken during the study helped so much. Compare the transmission of bacteria in an ICU with the % identified in this study.

To emphasize the epidemiology of the molecules of the identified bacterial species, they are practically not mentioned in the study. The ST must be determined for the isolates that were not performed.

How are the species related, for example, *Enterobacter cloacae* complex; plasmids are being transferred.

A conclusion is not identified at the end of the discussion and only that of the summary, which should be evaluated "This cohort study suggests that transmissions" transmission is low, we must take advantage of the analysis of genomes (molecular epidemiology, Inc groups of plasmids, mechanisms of resistance to β -lactams and other antibiotics, etc.).

Data availability: it must contain the access numbers to GenBank, review the data presented and it does not work correctly.

Reviewer #2 (Public repository details (Required)):

Accession numbers for genomic data were given but all of them were not accessible and were not consistent with the description in Methods.

Reviewer #2 (Comments for the Author):

Nurjadi et al. investigated risk factors and nosocomial transmission tracing of third-generation resistant Enterobacterales in a NICU in Germany. I read the manuscript with interests. Although this type of data are strongly limited (or biased) by the study setting of the hospital investigated, data described here is very important to understand nosocomial transmission and appropriate treatment strategies in the absence of similar reports. However, the manuscript contains descriptive errors/mistakes in manuscript which compromise the entire quality of the manuscript (especially microbiological descriptions). The authors need careful checking of the entire manuscript. I recommend the manuscript to be checked by microbiologists. Some technical methods used in genomic analysis are unclear or not compliant with the current standard thus need clarification/explanations (especially phylogenomic analysis).

Almost all third-generation cephalosporin resistant Enterobacterales isolates found in this study were species that have intrinsic AmpC resistance mechanisms without other acquired resistance mechanisms. Generally, these species, being part of gut microflora, are not considered as antimicrobial resistance organisms. Neonatal setting may slightly different from these understandings because nosocomial acquisition of these organisms can occur and can be threat to empirical treatment failure. The authors need to discuss to justify their study design. To support authors' design, I strongly recommend two points. First, include antimicrobial susceptibility data to describe these resistant organisms. Second, add outcomes of colonized/infected

patients vs. non-infected patients to define impacts of colonization infection by these organisms.

Title. Change Enterobacteriaceae to Enterobacterales, as the authors used in the text.

Line 30. Please clarify "cephalosporin" here is "third-generation cephalosporin". It is important to maintain consistency of the study objective, methods, results, and discussion. In Methods (line 30, 32), carbapenem-resistant Enterobacterales were investigated but the corresponding results were not shown. Running title is "MDRO" whereas third-generation cephalosporin resistance were used in other places. I suggest to limit the abstract to "third-generation cephalosporin resistance". Descriptions for CRE can be placed in the text.

Line 31. Did the authors use stool as a screening specimen? I think this is important and should be described in Abstract.

Line 37-38. There were other significant risk factors in the text. Why these two? Furthermore, the ORs here were different from those described in the text. I think here results from multivariate analysis (low birthweight and longer LOS) should be presented.

Line 77. Italicize "Staphylococcus aureus"

Line 106. Figure 1 does not describe risk factors.

Line 108. Results for CRE are missing. Antimicrobial susceptibility data should be described (the testing methods are described in Methods).

Line 115. In Figure 1, these 3 isolates which did not undergo WGS are not described.

Line 118. blaTEM-1 is not ESBL.

Line 122. "hqSNP" is not commonly used term. Needs definition and explanations. It is very unclear why the authors defined ≤ 15 SNPs as transmissions. Scientific evidence to support this is needed, especially in the setting that the authors did not use core SNP approach, which have been reported and validated in many studies.

Line 129. Do not italicize "complex".

Figure 3. Capitalize "K"lebsiella.

Line 131. Does this mean Enterobacter cloacae subsp. cloacae?

Line 134. AmpC should be "ampC" if this meant gene name.

Line 215. Missing dots between genus and species.

Line 254. Did the authors used the checklist for STROBE? At least title does not contain term indicating study design.

Line 271. Isolation and contact precautions were applied only when both cephalosporin-resistant and carbapenem-resistant organisms were detected?

Line 275. Please give a specific product name of medium.

Line 278. MALDI should be MALDI-TOF mass spectrometry.

Line 279. Please clarify if interpretive criteria for cephalosporin or carbapenem resistance changed during the study period.

Line 294. Please clarify criteria for species identification (cutoff value etc.) and supporting evidence, as this method is not the standard approach (ANI). Please disclose all reference genomes used.

Line 294-296. This approach is different from the current standard of the core SNP approach. Please briefly explain the differences and merits of choosing this approach with supporting evidence. Clarify reference genomes for each species and these should be included in the phylogenetic trees.

Line 308. The Bioproject number here is different from those described in Supplementary data. Furthermore, these numbers were not accessible so that I cannot assess the validity of genomic analysis.

Staff Comments:

Preparing Revision Guidelines

Please return the manuscript within 60 days; if you cannot complete the modification within this time period, please contact me. If you do not wish to modify the manuscript and prefer to submit it to another journal, please notify me of your decision immediately so that the manuscript may be formally withdrawn from consideration by Microbiology Spectrum.

Point-by-point response R1

Thank you for giving us the opportunity to revise and improve our manuscript. We have tried to the best of our ability to address the reviewers' comments and hope to have done so to your satisfaction.

Reviewer comments:

Reviewer #1 (Comments for the Author):

The study describes the characterization of colonizing isolates resistant to third-generation cephalosporins in a neonatal ICU.

Around the document they erroneously describe "third-generation cephalosporin Enterobacterales"

Response: this comment is unclear since the reviewer did not elaborate on why this term should be incorrect. In the study, we focused on third-generation cephalosporin and species belonging to the order Enterobacterales so that we are now aware of what "erroneous" meant in this context. If the reviewer refers to the term "*Enterobacteriaceae*" versus "Enterobacterales", this is now harmonized throughout the manuscript.

The title should be modified by eliminating the study period; including the country. That must be very well specified in M and M.

Response: we have now changed the title to "**Monocentric observational cohort study to investigate transmission of third-generation cephalosporin-resistant Enterobacterales in a neonatal intensive care unit, Heidelberg, Germany.**", following the recommendation of both reviewers.

The bacteria are isolated from the rectum of children; explaining colonization; however, whether there were infections in children is not described. It is not disputed how much it affected colonization and thus the results of the prevention measures taken during the study; in the same way the possible infections.

Response: we have now included the data on infections. There were 9 children with various infections, bloodstream infections, meningitis and wound infections. All of the 9 children with 3rd gen cephalosporin-resistant Enterobacterales were also colonized. This was not an intervention study or to evaluate the infection prevention measure so we would not speculate on the efficacy of specific measures with the small number of infected children in our setting. Furthermore, the infection prevention measures implemented are not targeted to prevent endogenous infection. Since the reviewer's question was unclear, we hope to have understood the question correctly and answered it adequately.

It is described that the identification by MALDI-TOF was used, and the results are not shown and if they agree with the species identified by WGS.

Response: the comparison of MALDI-TOF MS performance with species identification **was not the focus of the study**. It is generally known that the discriminatory power of MALDI-TOF MS is not as good as WGS. In general, the species identification was concordant. As expected, MALDI-TOF is limited in distinguishing the species level in the *Enterobacter cloacae* complex. However, this is generally acknowledged, and we believe that this is not relevant to the study. If the results were discordant (Gram-positive wrongly identified as Gram-negative), this would be omitted from the analysis.

The antimicrobial susceptibility obtained with VITEK2 does not correlate with the resistome obtained.

Response: the discrepancy between Vitek and genotypic resistance is also known and anticipated. Resistance mechanisms can be mediated by the presence or absence of a particular gene/resistance determinant but also be caused by overexpression of resistance determinants or efflux pumps. Therefore, it is plausible that phenotypic and genotypic resistance may not correlate.

The ideal way to identify bacterial species is the average nucleotide identity analysis; Therefore, it is suggested to use this program for the different species and correlate them with the MALDI-TOF result; Since if the MALDI-TOF database is not updated, it does not specifically identify the species of complexes.

Response: species identification based on mash screening, which is equivalent to ANI as it evaluates through both genomes the number of shared hashes and the overall nucleotide identity, was performed. This was not mentioned explicitly, but instead, we referred to our previous studies to avoid redundancy and due to word limitation. The different species of *Enterobacter cloacae* complex are elaborated in lines 135-138. With MALDI-TOF MS alone, definitive species designation is not possible, as correctly mentioned by the reviewer.

Introduction; detail more other studies from your hospital and what has changed.

Response: we disagree with this suggestion since we do not see any added benefits in listing previous studies done in our hospital. The relevant changes are mentioned already. Screening measures were implemented, but molecular typing was performed with PFGE or MLST. In this study, we implemented systematic genomic surveillance to incorporate strain typing using WGS into the routine screening. All relevant details were already included in the introduction (line 74-82).

Results.

Line 95-96, colonization or infection; it is not clear.

Response: this can be colonization or infection, or both. For the risk factors, detecting cephalosporin-resistant Enterobacterales refer to colonization and infection. We have replaced "or" with "and" for better clarity.

It would be necessary to describe and more to discuss the interesting data of figure 1. Why these species and how was it during the time?

Response: Figure 1 provides an overview of the species detected over the study period. The species are discussed in the respective subheadings in the results section. The species were not selected by any criteria, but these were all third-generation cephalosporin-resistant Enterobacterales. *Enterobacter hormaechei* was the most common, which could be considered a potential outbreak. We have added this sentence to lines 134-135 in the revised manuscript.

The Supplementary Figure S1 should be more described in results and correlate with ST and something that is not discussed and determined are the incompatibility groups of the plasmids. See an example. Duran-Bedolla J, Rodríguez-Medina N, Dunn M, Mosqueda-García D, Barrios-Camacho H, Aguilar-Vera A, Aguilar-Vera E, Suárez-Rodríguez R, Ramírez-Trujillo JA, Garza-Ramos U. Plasmids of the incompatibility group FIBK occurs in *Klebsiella variicola* from diverse ecological niches. *Int Microbiol.* 2023 Mar 27. doi: 10.1007/s10123-023-00346-0. Epub ahead of print. PMID: 36971854.

Response: Supplementary Figure 1 was meant to show a global distribution of the AMR genes between bacterial species. The Sequence Type and the plasmid incompatibility types are presented in Figures 2 and 3 for the most prevalent species. As we did not observe evidence of plasmid transfer between strains in those groups, we do not see the relevance of repeating this information in the supplementary Figure 1.

Little transmission of bacteria is identified, what is not clear is the phylogenetic relationship (clones) between the identified bacteria that were transmitted. This makes sense to the discussion that the measures taken during the study helped so much. Compare the transmission of bacteria in an ICU with the % identified in this study.

Response: we did not anticipate a lot of transmission since this study was performed in a non-outbreak setting. This is also the strength of the study, since most studies are performed only in outbreak settings, thus exaggerating the magnitude of transmission in "normal" settings. I am afraid that the reviewer misunderstood the aim of the study; this was a non-interventional study, so no effect on measures/interventions can be evaluated. This study was observational to see how many transmissions occur in a non-outbreak setting when systematic genomic surveillance is implemented. This study could deliver the necessary evidence to conduct an intervention study if transmission rates are too high and more rigorous infection prevention and control measures are needed.

To emphasize the epidemiology of the molecules of the identified bacterial species, they are practically not mentioned in the study. The ST must be determined for the isolates that were not performed.

Response: the MLST were provided in the supplementary table 1; the missing MLST were due to mutations in some loci and are now provided with the closest known ST, new ST or none when there is no scheme in Pubmlst. The phylogenetic relationship between AMR genes and the strains is already described in Figures 2-3 and supplementary Figure 1.

How are the species related, for example, *Enterobacter cloacae* complex; plasmids are being transferred

Response: the phylogenetic relationship between species is display on Suppl. Figure 1. We did not find any indication of plasmid transfer between species in our setting.

A conclusion is not identified at the end of the discussion and only that of the summary, which should be evaluated "This cohort study suggests that transmissions" transmission is low, we must take advantage of the analysis of genomes (molecular epidemiology, Inc groups of plasmids, mechanisms of resistance to b-lactams and other antibiotics, etc.).

Response: We disagree. The last paragraph is the conclusion. We do not explicitly state this paragraph as the conclusion since it is clear that this paragraph summarizes the findings, concluding that systematic molecular characterization can help detect transmission to guide infection prevention measures in this patient group.

Data availability: it must contain the access numbers to GenBank, review the data presented and it does not work correctly.

Response: The problem with the Bioproject was due to a Microsoft Excel mishap. All samples are in the Bioproject PRJNA954276, and somehow when copy pasting in Excel, it creates a sequence starting from PRJNA954276 to PRJNA954364. We apologize for this mistake, and it is now fixed. All biosample accession were correct. It is not currently possible to provide a reviewer link via NCBI, but if the reviewer wants to see the data, we could ask for an earlier release.

Reviewer #2 (Public repository details (Required)):

Response: see comment above.

Accession numbers for genomic data were given but all of them were not accessible and were not consistent with the description in Methods.

Response: see comment above.

Reviewer #2 (Comments for the Author):

Nurjadi et al. investigated risk factors and nosocomial transmission tracing of third-generation resistant Enterobacterales in a NICU in Germany. I read the manuscript with interests. Although this type of data are strongly limited (or biased) by the study setting of the hospital investigated, data described here is very important to understand nosocomial transmission and appropriate treatment strategies in the absence of similar reports. However, the manuscript contains descriptive errors/mistakes in manuscript which compromise the entire quality of the manuscript (especially microbiological descriptions). The authors need careful checking of the entire manuscript. I recommend the manuscript to be checked by microbiologists. Some technical methods used in genomic analysis are unclear or not compliant with the current standard thus need clarification/explanations (especially phylogenomic analysis).

Response: a clinical microbiologist is involved in the drafting and finalizing of the manuscript. Minor typos/formatting errors may happen and is not a rare phenomenon in manuscripts. We disagree that the minor deficits in the microbiological/methodological description due to space constraints question the manuscript's scientific integrity. To comply with the word limitations of many journals, we shortened the bioinformatics description since our group has published similar studies in the past, where the methods were described in great detail. We agree that we could have provided a more thorough description of the methods used in the manuscript and have now elaborated the methodology in the supplement.

Almost all third-generation cephalosporin resistant Enterobacterales isolates found in this study were species that have intrinsic AmpC resistance mechanisms without other acquired resistance mechanisms. Generally, these species, being part of gut microflora, are not considered as antimicrobial resistance organisms. Neonatal setting may slightly different from these understandings because nosocomial acquisition of these organisms can occur and can be threat to empirical treatment failure.

Response: from the microbiological perspective, this is true. However, according to the national guideline (commission for hospital hygiene and infection prevention, KRINKO, of the Robert Koch Institute in Germany), isolates with third-generation cephalosporin phenotypic resistance (the so-called 2-MRGN, multidrug-resistant gram-negatives with resistance towards broad-spectrum penicillin and third-generation cephalosporins) should be considered are relevant bacteria in these neonates (especially those with low-birth weight). The recommendations do not differentiate between mobile/transferable or intrinsic resistance mechanisms but are based solely on phenotypic resistance.

The authors need to discuss to justify their study design. To support authors' design, I strongly recommend two points. First, include antimicrobial susceptibility data to describe these resistant organisms.

Response: We think including all phenotypic susceptibility data is unnecessary. As you can see from the resistome, most of the bacterial species found in this patient group were multi-susceptible. Therefore, for clarity purposes, we only displayed the figures' genotypic resistance. The sequenced isolates were selected based on their resistance towards third-generation cephalosporin.

Second, add outcomes of colonized/infected patients vs. non-infected patients to define impacts of colonization infection by these organisms.

Response: we think it is unnecessary; colonized children are not necessarily sick, and we do not believe that assessing the clinical outcome based on colonization is clinically meaningful. In our cohort, only 9 children had an infection, all of which were colonized, so no meaningful statistical analysis could be performed comparing the colonized/infected and non-colonized/infected groups.

Title. Change Enterobacteriaceae to Enterobacterales, as the authors used in the text.

Response: changed

Line 30. Please clarify "cephalosporin" here is "third-generation cephalosporin". It is important to maintain consistency of the study objective, methods, results, and discussion. In Methods (line 30, 32), carbapenem-resistant Enterobacterales were investigated but the corresponding results were not shown. Running title is "MDRO" whereas third-generation cephalosporin resistance were used in other places. I suggest to limit the abstract to "third-generation cephalosporin resistance". Descriptions for CRE can be placed in the text.

Response: this was shortened to adhere to the word limitation of the abstract. We have now made minor alterations to address the suggestions/issues raised by the reviewer. We have removed CRE since this was not discussed and no data is presented in the manuscript regarding CRE.

Line 31. Did the authors use stool as a screening specimen? I think this is important and should be described in Abstract.

Response: screening was performed using rectal swabs as defined by the standard operating procedure of the local hospital hygiene team. We have added this to the abstract.

Line 37-38. There were other significant risk factors in the text. Why these two? Furthermore, the ORs here were

different from those described in the text. I think here results from multivariate analysis (low birthweight and longer LOS) should be presented.

Response: only significant risk factors from the multivariate analysis are presented in the abstract. We have now corrected this.

Line 77. Italicize "Staphylococcus aureus"

Response: done

Line 106. Figure 1 does not describe risk factors.

Response: we apologize for this error, this should have been Table 1.

Line 108. Results for CRE are missing. Antimicrobial susceptibility data should be described (the testing methods are described in Methods).

Response: the testing methods are described since this is the screening algorithm in our laboratory. Identification of isolates growing on the ESBL plate is confirmed by MALDI-TOF MS. AST was performed subsequently to confirm the phenotype. This algorithm was chosen to maximise detection sensitivity for third-generation cephalosporin-resistant Enterobacteriales and carbapenem-resistant Enterobacteriales. We have decided to omit the CRE results, as we have only found 2 CRE within the study period, which was not relevant to the message of the manuscript.

Line 115. In Figure 1, these 3 isolates which did not undergo WGS are not described.

Response: the figure 1 now include all the isolates. The three isolates were added and labelled with an asterisk to mark the non-recoverable isolates

Line 118. blaTEM-1 is not ESBL

Response: this was removed

Line 122. "hqSNP" is not commonly used term. Needs definition and explanations. It is very unclear why the authors defined ≤ 15 SNPs as transmissions. Scientific evidence to support this is needed, especially in the setting that the authors did not use core SNP approach, which have been reported and validated in many studies.

Response: We removed the term hqSNP for SNP for clarity. The transmission threshold was done by looking at the dispersion of the SNP distribution in each specie and cut at the first breakpoint. As this is an arbitrary measure, we now have to modify the methods to follow the methodology of Duvall et al. to estimate the threshold based on the date of isolation, genome size and mutation rates [1]. The new threshold is now in the manuscript. However, as no isolates were distant from another isolate between our arbitrary threshold and the new modeled threshold, it did not change our clustering and results.

Line 129. Do not italicize "complex".

Response: this is meant as a subheading. I will leave this decision to the typesetter, whether this should be italicized.

Figure 3. Capitalize "K"lebsiella.

Response: done

Line 131. Does this mean Enterobacter cloacae subsp. cloacae?

Response: yes, we have added subsp. to avoid misunderstanding.

Line 134. AmpC should be "ampC" if this meant gene name.

Response: corrected, the AmpC is referring to the protein.

Line 215. Missing dots between genus and species.

Response: this was due to formatting issues for various journals.

Line 254. Did the authors used the checklist for STROBE? At least title does not contain term indicating study design.

Response: yes, see lines 257-260 in the original and revised manuscript.

Line 271. Isolation and contact precautions were applied only when both cephalosporin-resistant and carbapenem-resistant organisms were detected?

Response: we apologize for this error. This was supposed to be "or". The detection of third-generation cephalosporin-resistant OR carbapenem-resistant Enterobacteriales would lead to isolation and contact precaution.

Line 275. Please give a specific product name of medium.

Response: the tradename of the medium was added.

Line 278. MALDI should be MALDI-TOF mass spectrometry.

Response: we have corrected this

Line 279. Please clarify if interpretive criteria for cephalosporin or carbapenem resistance changed during the study period.

Response: the interpretation for AST (S, I, R category) changed in 2019 but since only “R”. There were, however, no change in category for isolates tested in 2018. We have added this into the text/methods section.

Line 294. Please clarify criteria for species identification (cutoff value etc.) and supporting evidence, as this method is not the standard approach (ANI). Please disclose all reference genomes used.

Response: species identification was done using mash screen amongst a database of representative genomes—all identity thresholds were higher than 96%. For space reasons, we referred to older publications, where we described the methodology more elaborately. We have now added this detail in the manuscript and the identity in supplementary table 1.

Line 294-296. This approach is different from the current standard of the core SNP approach. Please briefly explain the differences and merits of choosing this approach with supporting evidence. Clarify reference genomes for each species and these should be included in the phylogenetic trees.

Response: We disagree with the reviewer on that point. Gubbins is now becoming more and more used to cope with horizontal gene transfer, recombination hotspots which can inflate the SNP distance between strains in short/mid-term evolution setting. Using raw SNP approach is actually only the first step for epidemiological analysis and recombination corrected phylogeny showed better resolution [2]

Line 308. The Bioproject number here is different from those described in Supplementary data. Furthermore, these numbers were not accessible so that I cannot assess the validity of genomic analysis.

Response: this was corrected.

References

1. Duval A, Opatowski L, Brisse S. Defining genomic epidemiology thresholds for common-source bacterial outbreaks: a modelling study. *Lancet Microbe*. 2023;4(5):e349-e57.
2. Didelot X, Parkhill J. A scalable analytical approach from bacterial genomes to epidemiology. *Philos Trans R Soc Lond B Biol Sci*. 2022;377(1861):20210246.

July 31, 2023

Prof. Dennis Nurjadi
Universität zu Lubeck
Department of Infectious Diseases and Microbiology
Ratzeburger Allee 160 Haus 50
Lübeck 23538
Germany

Re: Spectrum02038-23R1 (Monocentric observational cohort study to investigate the transmission of third-generation cephalosporin-resistant Enterobacterales in a neonatal intensive care unit, Heidelberg, Germany.)

Dear Prof. Dennis Nurjadi:

Thank you for submitting your manuscript to Microbiology Spectrum. As you will see your paper is very close to acceptance. Please modify the manuscript along the lines I have recommended. As these revisions are quite minor, I expect that you should be able to turn in the revised paper in less than 30 days, if not sooner. If your manuscript was reviewed, you will find the reviewers' comments below.

When submitting the revised version of your paper, please provide (1) point-by-point responses to the issues raised by the reviewers as file type "Response to Reviewers," not in your cover letter, and (2) a PDF file that indicates the changes from the original submission (by highlighting or underlining the changes) as file type "Marked Up Manuscript - For Review Only". Please use this link to submit your revised manuscript. Detailed instructions on submitting your revised paper are below.

Link Not Available

Sincerely,

Katharina Schaufler

Editor comments:

Please change third-generation cephalosporin Enterobacterales to third-generation cephalosporin-RESISTANT Enterobacterales/rephrase third-generation cephalosporin colonization.

Preparing Revision Guidelines

Please return the manuscript within 60 days; if you cannot complete the modification within this time period, please contact me. If you do not wish to modify the manuscript and prefer to submit it to another journal, please notify me of your decision immediately so that the manuscript may be formally withdrawn from consideration by Microbiology Spectrum.

Point-by-point response R1

Thank you for giving us the opportunity to revise and improve our manuscript. We have tried to the best of our ability to address the reviewers' comments and hope to have done so to your satisfaction.

Reviewer comments:

Reviewer #1 (Comments for the Author):

The study describes the characterization of colonizing isolates resistant to third-generation cephalosporins in a neonatal ICU.

Around the document they erroneously describe "third-generation cephalosporin Enterobacterales"

Response: this comment is unclear since the reviewer did not elaborate on why this term should be incorrect. In the study, we focused on third-generation cephalosporin and species belonging to the order Enterobacterales so that we are now aware of what "erroneous" meant in this context. If the reviewer refers to the term "*Enterobacteriaceae*" versus "Enterobacterales", this is now harmonized throughout the manuscript.

The title should be modified by eliminating the study period; including the country. That must be very well specified in M and M.

Response: we have now changed the title to "**Monocentric observational cohort study to investigate transmission of third-generation cephalosporin-resistant Enterobacterales in a neonatal intensive care unit, Heidelberg, Germany.**", following the recommendation of both reviewers.

The bacteria are isolated from the rectum of children; explaining colonization; however, whether there were infections in children is not described. It is not disputed how much it affected colonization and thus the results of the prevention measures taken during the study; in the same way the possible infections.

Response: we have now included the data on infections. There were 9 children with various infections, bloodstream infections, meningitis and wound infections. All of the 9 children with 3rd gen cephalosporin-resistant Enterobacterales were also colonized. This was not an intervention study or to evaluate the infection prevention measure so we would not speculate on the efficacy of specific measures with the small number of infected children in our setting. Furthermore, the infection prevention measures implemented are not targeted to prevent endogenous infection. Since the reviewer's question was unclear, we hope to have understood the question correctly and answered it adequately.

It is described that the identification by MALDI-TOF was used, and the results are not shown and if they agree with the species identified by WGS.

Response: the comparison of MALDI-TOF MS performance with species identification **was not the focus of the study**. It is generally known that the discriminatory power of MALDI-TOF MS is not as good as WGS. In general, the species identification was concordant. As expected, MALDI-TOF is limited in distinguishing the species level in the *Enterobacter cloacae* complex. However, this is generally acknowledged, and we believe that this is not relevant to the study. If the results were discordant (Gram-positive wrongly identified as Gram-negative), this would be omitted from the analysis.

The antimicrobial susceptibility obtained with VITEK2 does not correlate with the resistome obtained.

Response: the discrepancy between Vitek and genotypic resistance is also known and anticipated. Resistance mechanisms can be mediated by the presence or absence of a particular gene/resistance determinant but also be caused by overexpression of resistance determinants or efflux pumps. Therefore, it is plausible that phenotypic and genotypic resistance may not correlate.

The ideal way to identify bacterial species is the average nucleotide identity analysis; Therefore, it is suggested to use this program for the different species and correlate them with the MALDI-TOF result; Since if the MALDI-TOF database is not updated, it does not specifically identify the species of complexes.

Response: species identification based on mash screening, which is equivalent to ANI as it evaluates through both genomes the number of shared hashes and the overall nucleotide identity, was performed. This was not mentioned explicitly, but instead, we referred to our previous studies to avoid redundancy and due to word limitation. The different species of *Enterobacter cloacae* complex are elaborated in lines 135-138. With MALDI-TOF MS alone, definitive species designation is not possible, as correctly mentioned by the reviewer.

Introduction; detail more other studies from your hospital and what has changed.

Response: we disagree with this suggestion since we do not see any added benefits in listing previous studies done in our hospital. The relevant changes are mentioned already. Screening measures were implemented, but molecular typing was performed with PFGE or MLST. In this study, we implemented systematic genomic surveillance to incorporate strain typing using WGS into the routine screening. All relevant details were already included in the introduction (line 74-82).

Results.

Line 95-96, colonization or infection; it is not clear.

Response: this can be colonization or infection, or both. For the risk factors, detecting cephalosporin-resistant Enterobacterales refer to colonization and infection. We have replaced "or" with "and" for better clarity.

It would be necessary to describe and more to discuss the interesting data of figure 1. Why these species and how was it during the time?

Response: Figure 1 provides an overview of the species detected over the study period. The species are discussed in the respective subheadings in the results section. The species were not selected by any criteria, but these were all third-generation cephalosporin-resistant Enterobacterales. *Enterobacter hormaechei* was the most common, which could be considered a potential outbreak. We have added this sentence to lines 134-135 in the revised manuscript.

The Supplementary Figure S1 should be more described in results and correlate with ST and something that is not discussed and determined are the incompatibility groups of the plasmids. See an example. Duran-Bedolla J, Rodríguez-Medina N, Dunn M, Mosqueda-García D, Barrios-Camacho H, Aguilar-Vera A, Aguilar-Vera E, Suárez-Rodríguez R, Ramírez-Trujillo JA, Garza-Ramos U. Plasmids of the incompatibility group FIBK occurs in *Klebsiella variicola* from diverse ecological niches. *Int Microbiol.* 2023 Mar 27. doi: 10.1007/s10123-023-00346-0. Epub ahead of print. PMID: 36971854.

Response: Supplementary Figure 1 was meant to show a global distribution of the AMR genes between bacterial species. The Sequence Type and the plasmid incompatibility types are presented in Figures 2 and 3 for the most prevalent species. As we did not observe evidence of plasmid transfer between strains in those groups, we do not see the relevance of repeating this information in the supplementary Figure 1.

Little transmission of bacteria is identified, what is not clear is the phylogenetic relationship (clones) between the identified bacteria that were transmitted. This makes sense to the discussion that the measures taken during the study helped so much. Compare the transmission of bacteria in an ICU with the % identified in this study.

Response: we did not anticipate a lot of transmission since this study was performed in a non-outbreak setting. This is also the strength of the study, since most studies are performed only in outbreak settings, thus exaggerating the magnitude of transmission in "normal" settings. I am afraid that the reviewer misunderstood the aim of the study; this was a non-interventional study, so no effect on measures/interventions can be evaluated. This study was observational to see how many transmissions occur in a non-outbreak setting when systematic genomic surveillance is implemented. This study could deliver the necessary evidence to conduct an intervention study if transmission rates are too high and more rigorous infection prevention and control measures are needed.

To emphasize the epidemiology of the molecules of the identified bacterial species, they are practically not mentioned in the study. The ST must be determined for the isolates that were not performed.

Response: the MLST were provided in the supplementary table 1; the missing MLST were due to mutations in some loci and are now provided with the closest known ST, new ST or none when there is no scheme in Pubmlst. The phylogenetic relationship between AMR genes and the strains is already described in Figures 2-3 and supplementary Figure 1.

How are the species related, for example, *Enterobacter cloacae* complex; plasmids are being transferred

Response: the phylogenetic relationship between species is display on Suppl. Figure 1. We did not find any indication of plasmid transfer between species in our setting.

A conclusion is not identified at the end of the discussion and only that of the summary, which should be evaluated "This cohort study suggests that transmissions" transmission is low, we must take advantage of the analysis of genomes (molecular epidemiology, Inc groups of plasmids, mechanisms of resistance to b-lactams and other antibiotics, etc.).

Response: We disagree. The last paragraph is the conclusion. We do not explicitly state this paragraph as the conclusion since it is clear that this paragraph summarizes the findings, concluding that systematic molecular characterization can help detect transmission to guide infection prevention measures in this patient group.

Data availability: it must contain the access numbers to GenBank, review the data presented and it does not work correctly.

Response: The problem with the Bioproject was due to a Microsoft Excel mishap. All samples are in the Bioproject PRJNA954276, and somehow when copy pasting in Excel, it creates a sequence starting from PRJNA954276 to PRJNA954364. We apologize for this mistake, and it is now fixed. All biosample accession were correct. It is not currently possible to provide a reviewer link via NCBI, but if the reviewer wants to see the data, we could ask for an earlier release.

Reviewer #2 (Public repository details (Required)):

Response: see comment above.

Accession numbers for genomic data were given but all of them were not accessible and were not consistent with the description in Methods.

Response: see comment above.

Reviewer #2 (Comments for the Author):

Nurjadi et al. investigated risk factors and nosocomial transmission tracing of third-generation resistant Enterobacterales in a NICU in Germany. I read the manuscript with interests. Although this type of data are strongly limited (or biased) by the study setting of the hospital investigated, data described here is very important to understand nosocomial transmission and appropriate treatment strategies in the absence of similar reports. However, the manuscript contains descriptive errors/mistakes in manuscript which compromise the entire quality of the manuscript (especially microbiological descriptions). The authors need careful checking of the entire manuscript. I recommend the manuscript to be checked by microbiologists. Some technical methods used in genomic analysis are unclear or not compliant with the current standard thus need clarification/explanations (especially phylogenomic analysis).

Response: a clinical microbiologist is involved in the drafting and finalizing of the manuscript. Minor typos/formatting errors may happen and is not a rare phenomenon in manuscripts. We disagree that the minor deficits in the microbiological/methodological description due to space constraints question the manuscript's scientific integrity. To comply with the word limitations of many journals, we shortened the bioinformatics description since our group has published similar studies in the past, where the methods were described in great detail. We agree that we could have provided a more thorough description of the methods used in the manuscript and have now elaborated the methodology in the supplement.

Almost all third-generation cephalosporin resistant Enterobacterales isolates found in this study were species that have intrinsic AmpC resistance mechanisms without other acquired resistance mechanisms. Generally, these species, being part of gut microflora, are not considered as antimicrobial resistance organisms. Neonatal setting may slightly different from these understandings because nosocomial acquisition of these organisms can occur and can be threat to empirical treatment failure.

Response: from the microbiological perspective, this is true. However, according to the national guideline (commission for hospital hygiene and infection prevention, KRINKO, of the Robert Koch Institute in Germany), isolates with third-generation cephalosporin phenotypic resistance (the so-called 2-MRGN, multidrug-resistant gram-negatives with resistance towards broad-spectrum penicillin and third-generation cephalosporins) should be considered are relevant bacteria in these neonates (especially those with low-birth weight). The recommendations do not differentiate between mobile/transferable or intrinsic resistance mechanisms but are based solely on phenotypic resistance.

The authors need to discuss to justify their study design. To support authors' design, I strongly recommend two points. First, include antimicrobial susceptibility data to describe these resistant organisms.

Response: We think including all phenotypic susceptibility data is unnecessary. As you can see from the resistome, most of the bacterial species found in this patient group were multi-susceptible. Therefore, for clarity purposes, we only displayed the figures' genotypic resistance. The sequenced isolates were selected based on their resistance towards third-generation cephalosporin.

Second, add outcomes of colonized/infected patients vs. non-infected patients to define impacts of colonization infection by these organisms.

Response: we think it is unnecessary; colonized children are not necessarily sick, and we do not believe that assessing the clinical outcome based on colonization is clinically meaningful. In our cohort, only 9 children had an infection, all of which were colonized, so no meaningful statistical analysis could be performed comparing the colonized/infected and non-colonized/infected groups.

Title. Change Enterobacteriaceae to Enterobacterales, as the authors used in the text.

Response: changed

Line 30. Please clarify "cephalosporin" here is "third-generation cephalosporin". It is important to maintain consistency of the study objective, methods, results, and discussion. In Methods (line 30, 32), carbapenem-resistant Enterobacterales were investigated but the corresponding results were not shown. Running title is "MDRO" whereas third-generation cephalosporin resistance were used in other places. I suggest to limit the abstract to "third-generation cephalosporin resistance". Descriptions for CRE can be placed in the text.

Response: this was shortened to adhere to the word limitation of the abstract. We have now made minor alterations to address the suggestions/issues raised by the reviewer. We have removed CRE since this was not discussed and no data is presented in the manuscript regarding CRE.

Line 31. Did the authors use stool as a screening specimen? I think this is important and should be described in Abstract.

Response: screening was performed using rectal swabs as defined by the standard operating procedure of the local hospital hygiene team. We have added this to the abstract.

Line 37-38. There were other significant risk factors in the text. Why these two? Furthermore, the ORs here were

different from those described in the text. I think here results from multivariate analysis (low birthweight and longer LOS) should be presented.

Response: only significant risk factors from the multivariate analysis are presented in the abstract. We have now corrected this.

Line 77. Italicize "Staphylococcus aureus"

Response: done

Line 106. Figure 1 does not describe risk factors.

Response: we apologize for this error, this should have been Table 1.

Line 108. Results for CRE are missing. Antimicrobial susceptibility data should be described (the testing methods are described in Methods).

Response: the testing methods are described since this is the screening algorithm in our laboratory. Identification of isolates growing on the ESBL plate is confirmed by MALDI-TOF MS. AST was performed subsequently to confirm the phenotype. This algorithm was chosen to maximise detection sensitivity for third-generation cephalosporin-resistant Enterobacterales and carbapenem-resistant Enterobacterales. We have decided to omit the CRE results, as we have only found 2 CRE within the study period, which was not relevant to the message of the manuscript.

Line 115. In Figure 1, these 3 isolates which did not undergo WGS are not described.

Response: the figure 1 now include all the isolates. The three isolates were added and labelled with an asterisk to mark the non-recoverable isolates

Line 118. blaTEM-1 is not ESBL

Response: this was removed

Line 122. "hqSNP" is not commonly used term. Needs definition and explanations. It is very unclear why the authors defined ≤ 15 SNPs as transmissions. Scientific evidence to support this is needed, especially in the setting that the authors did not use core SNP approach, which have been reported and validated in many studies.

Response: We removed the term hqSNP for SNP for clarity. The transmission threshold was done by looking at the dispersion of the SNP distribution in each specie and cut at the first breakpoint. As this is an arbitrary measure, we now have to modify the methods to follow the methodology of Duvall et al. to estimate the threshold based on the date of isolation, genome size and mutation rates [1]. The new threshold is now in the manuscript. However, as no isolates were distant from another isolate between our arbitrary threshold and the new modeled threshold, it did not change our clustering and results.

Line 129. Do not italicize "complex".

Response: this is meant as a subheading. I will leave this decision to the typesetter, whether this should be italicized.

Figure 3. Capitalize "K"lebsiella.

Response: done

Line 131. Does this mean Enterobacter cloacae subsp. cloacae?

Response: yes, we have added subsp. to avoid misunderstanding.

Line 134. AmpC should be "ampC" if this meant gene name.

Response: corrected, the AmpC is referring to the protein.

Line 215. Missing dots between genus and species.

Response: this was due to formatting issues for various journals.

Line 254. Did the authors used the checklist for STROBE? At least title does not contain term indicating study design.

Response: yes, see lines 257-260 in the original and revised manuscript.

Line 271. Isolation and contact precautions were applied only when both cephalosporin-resistant and carbapenem-resistant organisms were detected?

Response: we apologize for this error. This was supposed to be "or". The detection of third-generation cephalosporin-resistant OR carbapenem-resistant Enterobacterales would lead to isolation and contact precaution.

Line 275. Please give a specific product name of medium.

Response: the tradename of the medium was added.

Line 278. MALDI should be MALDI-TOF mass spectrometry.

Response: we have corrected this

Line 279. Please clarify if interpretive criteria for cephalosporin or carbapenem resistance changed during the study period.

Response: the interpretation for AST (S, I, R category) changed in 2019 but since only “R”. There were, however, no change in category for isolates tested in 2018. We have added this into the text/methods section.

Line 294. Please clarify criteria for species identification (cutoff value etc.) and supporting evidence, as this method is not the standard approach (ANI). Please disclose all reference genomes used.

Response: species identification was done using mash screen amongst a database of representative genomes—all identity thresholds were higher than 96%. For space reasons, we referred to older publications, where we described the methodology more elaborately. We have now added this detail in the manuscript and the identity in supplementary table 1.

Line 294-296. This approach is different from the current standard of the core SNP approach. Please briefly explain the differences and merits of choosing this approach with supporting evidence. Clarify reference genomes for each species and these should be included in the phylogenetic trees.

Response: We disagree with the reviewer on that point. Gubbins is now becoming more and more used to cope with horizontal gene transfer, recombination hotspots which can inflate the SNP distance between strains in short/mid-term evolution setting. Using raw SNP approach is actually only the first step for epidemiological analysis and recombination corrected phylogeny showed better resolution [2]

Line 308. The Bioproject number here is different from those described in Supplementary data. Furthermore, these numbers were not accessible so that I cannot assess the validity of genomic analysis.

Response: this was corrected.

References

1. Duval A, Opatowski L, Brisse S. Defining genomic epidemiology thresholds for common-source bacterial outbreaks: a modelling study. *Lancet Microbe*. 2023;4(5):e349-e57.
2. Didelot X, Parkhill J. A scalable analytical approach from bacterial genomes to epidemiology. *Philos Trans R Soc Lond B Biol Sci*. 2022;377(1861):20210246.

August 4, 2023

Prof. Dennis Nurjadi
Universität zu Lubeck
Department of Infectious Diseases and Microbiology
Ratzeburger Allee 160 Haus 50
Lübeck 23538
Germany

Re: Spectrum02038-23R2 (Monocentric observational cohort study to investigate the transmission of third-generation cephalosporin-resistant Enterobacterales in a neonatal intensive care unit, Heidelberg, Germany.)

Dear Prof. Dennis Nurjadi:

Your manuscript has been accepted, and I am forwarding it to the ASM Journals Department for publication. You will be notified when your proofs are ready to be viewed.

Sincerely,

Katharina Schaufler
Editor, Microbiology Spectrum
